

# Osteokines and the vasculature: a review of the *in vitro* effects of osteocalcin, fibroblast growth factor-23 and lipocalin-2

Sophie A. Millar, Susan I. Anderson and Saoirse E. O'Sullivan

Division of Graduate Entry Medicine and Medical Sciences, School of Medicine, Royal Derby Hospital, University of Nottingham, Derby, United Kingdom

## ABSTRACT

Bone-derived factors that demonstrate extra-skeletal functions, also termed osteokines, are fast becoming a highly interesting and focused area of cross-disciplinary endocrine research. Osteocalcin (OCN), fibroblast growth factor-23 (FGF23) and lipocalin-2 (LCN-2), produced in bone, comprise an important endocrine system that is finely tuned with other organs to ensure homeostatic balance and health. This review aims to evaluate *in vitro* evidence of the direct involvement of these proteins in vascular cells and whether any causal roles in cardiovascular disease or inflammation can be supported. PubMed, Medline, Embase and Google Scholar were searched for relevant research articles investigating the exogenous addition of OCN, FGF23 or LCN-2 to vascular smooth muscle or endothelial cells. Overall, these osteokines are directly vasoactive across a range of human and animal vascular cells. Both OCN and FGF23 have anti-apoptotic properties and increase eNOS phosphorylation and nitric oxide production through Akt signalling in human endothelial cells. OCN improves intracellular insulin signalling and demonstrates protective effects against endoplasmic reticulum stress in murine and human endothelial cells. OCN may be involved in calcification but further research is warranted, while there is no evidence for a pro-calcific effect of FGF23 *in vitro*. FGF23 and LCN-2 increase proliferation in some cell types and increase and decrease reactive oxygen species generation, respectively. LCN-2 also has anti-apoptotic effects but may increase endoplasmic reticulum stress as well as have pro-inflammatory and pro-angiogenic properties in human vascular endothelial and smooth muscle cells. There is no strong evidence to support a pathological role of OCN or FGF23 in the vasculature based on these findings. In contrast, they may in fact support normal endothelial functioning, vascular homeostasis and vasodilation. No studies examined whether OCN or FGF23 may have a role in vascular inflammation. Limited studies with LCN-2 indicate a pro-inflammatory and possible pathological role in the vasculature but further mechanistic data is required. Overall, these osteokines pose intriguing functions which should be investigated comprehensively to assess their relevance to cardiovascular disease and health in humans.

Corresponding author
Sophie A. Millar,
stxsamil@nottingham.ac.uk

## INTRODUCTION

The role of the skeleton has greatly evolved from its premise of being a functional tissue providing primary protection, support, and haematopoietic maintenance. It is now recognised as an endocrine organ providing exciting opportunities for cross-disciplinary research. One candidate thread of enquiry is the skeleton's interaction with the vascular system. Bone is a highly vascularised tissue, which is important for growth, remodelling and repair. The endothelial cells lining blood vessels have an important influence on bone cells, often in a paracrine manner, establishing a 'bone-vascular axis' (*Brandi & Collin-Osdoby, 2006*). This traditionally viewed axis is also evolving to reflect that on the other hand, secreted, circulating bone-derived factors have direct effects on vascular cells themselves.

Recent research has demonstrated that three 'osteokines', namely osteocalcin (OCN), fibroblast growth factor-23 (FGF23) and lipocalin-2 (LCN-2), comprise an important endocrine system that is finely tuned with other organs to ensure homeostatic balance and health (*Han et al., 2018*). The endocrine functions of OCN have been extensively reviewed elsewhere (*Ducy, 2011*; *Li et al., 2016*; *Karsenty, 2017*; *Han et al., 2018*). Briefly, OCN has been reported to play an active role in the regulation of energy metabolism by improving glucose tolerance and insulin sensitivity and insulin production (*Lee et al., 2007*; *Oury et al., 2011*; *Oury et al., 2013*). Elsewhere, $OCN^{-/-}$ male mice were observed to be poor breeders due to decreased testosterone production and thus has been linked to fertility (*Oury et al., 2011*). Interestingly, these $OCN^{-/-}$ mice also displayed behaviours associated with anxiety and depression and it was demonstrated that OCN can cross the blood–brain-barrier and enhance monoamine neurotransmitter synthesis, drawing a link between OCN and cognition (*Oury et al., 2013*). FGF23 is well characterised as the regulator of phosphate, parathyroid hormone (PTH), and vitamin D in the body which has been recently reviewed elsewhere (*Richter & Faul, 2018*). Briefly, in the kidneys, FGF23 increases phosphate excretion in urine, thus lowering serum concentrations. FGF23 is secreted by bone following stimulation by increased circulating vitamin D and extracellular phosphate. FGF23 in turn also supresses $1,25(OH)_2D_3$ production. Thus, a regulatory feedback loop exists between the kidneys and bone through FGF23. In the kidneys, FGF23 also increases reabsorption of calcium and sodium (*Andrukhova et al., 2014a*; *Andrukhova et al., 2014b*), while in the parathyroid gland FGF23 supresses PTH production. The endocrine role of LCN-2 was an interesting discovery through the use of mice lacking LCN-2 in osteoblasts ($Lcn2^{osb-/-}$). These mice displayed a decrease in insulin sensitivity and secretion, through direct activity on pancreatic β-cells, and decreased glucose tolerance (*Mosialou et al., 2017*). It has also been shown that LCN-2 crosses the blood brain barrier and activates melanocortin 4 receptor-dependent appetite signalling in the hypothalamus (*Mosialou et al., 2017*).

While these investigations have drawn evidence for the skeletal control of energy metabolism, mineral homeostasis and much more, the effects of these hormones on the vasculature itself has also gathered interest. Conflicting observational data exists between OCN, FGF23 and LCN-2 and markers of cardiovascular disease in humans (*Dalal et al., 2011*; *Wang, 2012*; *Millar et al., 2017*), which are discussed further within this review.

In order to assess the relevance of these osteokines to cardiovascular health, we aimed to provide an analysis of the direct effects of OCN, FGF23 and LCN-2 on micro- and macro-vascular cells *in vitro*, and to highlight gaps in the literature which need addressing, for use by clinicians and researchers.

## SURVEY METHODOLOGY

To retrieve articles concerning the exogenous administration of OCN, FGF23 and LCN-2 to micro- or macro-vascular cells, the following search terms were used for database searching (PubMed, Embase, Medline, Google Scholar): fibroblast growth factor 23, bone gamma-carboxyglutamic acid protein, osteocalcin, lipocalin 2, LCN-2, neutrophil gelatinase-associated lipocalin, NGAL, vascular cell, vascular endothelial cell, and vascular smooth muscle cell. Additionally, the reference lists of relevant review articles and included articles were hand-searched for further studies. Searches were carried out up until 20/09/2018. Retrieved articles were included if they described the exogenous addition of OCN, FGF23 or LCN-2 to vascular cells *in vitro* i.e., studies that examined the endogenous over-expression of these osteokines were excluded, as the results would not be deemed to be bone-derived. Similarly, passive reporting of osteokines e.g., expression studies, were not evaluated as this approach excludes the possibility of bone-derived effects and we were interested in clarifying any direct, causal effects of 'circulating' osteokines. Only peer-reviewed, full-text articles were included, not conference abstracts or articles not in English.

## OSTEOCALCIN

Osteocalcin (OCN) has been pioneered as a skeletal hormone for over a decade now, solidifying a role in influencing energy metabolism (*Lee et al., 2007*). The mature OCN is a 46 or 49 amino acid protein in mice and humans respectively, predominantly produced by osteoblasts, and is the most abundant, non-collagenous protein in the mineralised matrix of bone (*Hauschka, Lian & Gallop, 1975*; *Price et al., 1976*; *Celeste et al., 1986*; *Hauschka et al., 1989*). Due to the presence of 3 glutamic acid residues which may undergo carboxylation, OCN is present in various forms. These include undercarboxylated OCN with 1 or 2 glutamic acid residues (unOCN), carboxylated OCN with no glutamic acid residues (cOCN), or uncarboxylated OCN (ucOCN) with 3 glutamic acid residues. Carboxylation induces a conformational change resulting in a high affinity for calcium ions. Thus, OCN is also referred to as bone γ-carboxyglutamic acid-containing protein (BGLAP) and binds to hydroxyapatite crystal lattices present in bone extra-cellular matrix (*Hauschka et al., 1989*). unOCN and ucOCN have less affinity to hydroxyapatite and are more readily released into the circulation. These forms have been the most widely studied and are considered to be the 'active' forms of OCN (*Plantalech et al., 1991*; *Cairns & Price, 1994*; *Ferron et al., 2010*). Despite the abundance of OCN in bone, its exact role and function in bone reformation has not been clearly defined. However, serum concentrations of OCN may be used as a marker of bone turnover as it is widely accepted to be released from mature osteoblasts during bone formation (*Bellows, Reimers & Heersche, 1999*).
## Circulating concentrations of OCN

There is currently no consensus on which OCN fragments to measure, and there are no international or national standards (*Lee, Hodges & Eastell, 2000*). Total OCN concentrations have been reported by Hannemann et al. by age and gender (*Hannemann et al., 2013*). However, reported serum ucOCN and unOCN concentrations in studies have largely varied, for example between 0.1 and 0.3 ng/mL, and up to 21.4 ng/mL or 33.0 ng/mL (*Plantalech et al., 1991*; *Binkley et al., 2000*; *Iki et al., 2012*). OCN concentrations increase with age but can be reduced by supplementation with vitamin K (*Knapen, Hamulyak & Vermeer, 1989*; *Plantalech et al., 1991*; *Knapen et al., 1993*). cOCN concentrations have also not been defined and have been reported to lie around 8–10 ng/mL or higher e.g., 23.9 ng/mL (*Luukinen et al., 2000*; *Tsukamoto et al., 2000*; *Sugiyama & Kawai, 2001*). To add to the complexity of measuring circulating OCN, only 25% is found intact, and further additional forms are also present including N-terminal, mid-region, mid-region-C-terminal, and C-terminal fragments released by proteolytic cleavage (*Garnero et al., 1994*).

## OCN and the vasculature

A number of cross-sectional studies have highlighted associations between OCN concentrations and atherosclerosis and calcification, however, we found that a meta-analysis of these data could not provide a conclusive role as study results were mixed and conflicting with a number of limitations (*Millar et al., 2017*). Analysing the direct interaction of OCN and its forms with vascular cells may elucidate whether OCN is a mediator or marker of vascular calcification or atherosclerosis, and whether it functions to affect vascular cells independently of its previously noted metabolic influences. Cross-sectional evidence also exists regarding a role for OCN in inflammation, with some population data highlighting an inverse association between OCN concentrations and systemic inflammatory makers such as C-reactive protein and interleukin-6 (*Pittas et al., 2009*; *Chen et al., 2013*; *Lucey et al., 2013*; *Sarkar & Choudhury, 2013*; *Liao et al., 2015*). A recent summary of *in vivo* ucOCN treatments in mice and rats concluded that ucOCN protects vascular function and protects against markers that are commonly associated with the development of atherosclerosis (*Tacey et al., 2018*). However, these findings are concomitant with improved metabolic status, and as such, the establishment of direct effects of OCN and its forms on vascular cells needs confirmation. The following section aims to summarise *in vitro* investigations on the exogenous addition of OCN to vascular endothelial or smooth muscle cells (see Table 1).

### OCN in human and animal cells

*Jung et al. (2013)* reported an increase in Akt and eNOS phosphorylation and nitric oxide (NO) production in a PI3-kinase dependent manner in human aortic endothelial cells (HAECs) when treated with ucOCN (0.3–30.0 ng/mL) (*Jung et al., 2013*). They also demonstrated an anti-apoptotic effect of ucOCN, in which 30 ng/mL pre-treatment prevented linoleic acid induced apoptosis, also via the PI3K/Akt pathway. Similarly, another study showed an increase in eNOS phosphorylation in HAECs after 30 min incubation with ucOCN (25 and 100 ng/mL) (*Kondo et al., 2016*). Interestingly, eNOS phosphorylation

**Table 1  Summary of studies investigating *in vitro* effects of osteocalcin on human and animal vascular cells.**

| Study | Cell type | Type of OCN | Concentration | Results | Conclusions |
|---|---|---|---|---|---|
| *Idelevich, Rais & Monsonego-Ornan (2011)* | Murine VSMCs | Total OCN | Unknown | OCN was shown to be a glucose metabolism-modulating factor, through HIF-1α. | OCN is involved in glucose-metabolism |
| *Zhou et al. (2013)* | Murine VECs and VSMCs | ucOCN | 5 ng/mL | ucOCN protected against tunicamycin induced ER stress and autophagy, and improved insulin signalling in an Akt/mTOR/NFκB pathway. | Protective effect of ucOCN and improved insulin signalling |
| *Jung et al. (2013)* | Human AECs | ucOCN | 0.3–30 ng/mL | Increased eNOS and NO, and prevented linoleic acid induced apoptosis in PI3-K/Akt dependent manner. | Protective effect of ucOCN |
| *Dou et al. (2014)* | HUVECs | Total OCN | 10–150 ng/mL | >30 ng/mL OCN increased eNOS and Akt phosphorylation in a time- and dose-dependent manner. | Potential protective effect of total OCN |
| *Kondo et al. (2016)* | Human AECs | ucOCN cOCN | 25 and 100 ng/mL | ucOCN increased eNOS phosphorylation after 30 min but cOCN had no effect. | Potential protective effect of ucOCN |
| *Guo et al. (2017)* | HUVECs | ucOCN | 5 ng/mL | In insulin-resistant cells (induced by tunicamycin) ucOCN improved insulin signal transduction via PI3-K/Akt/NFκB pathways. | ucOCN is involved in insulin signalling |

**Notes.**

Abbreviations: OCN, osteocalcin; VECs, vascular endothelial cells; VSMCs, vascular smooth muscle cells; AECs, aortic endothelial cells; HUVECs, human umbilical vein endothelial cells; ucOCN, uncarboxylated OCN; cOCN, carboxylated OCN; ER, endoplasmic reticulum; mTOR, mechanistic target of rapamycin; NFκB, Nuclear factor-κB; HIF-1α, hypoxia inducible factor 1−α; PI3-K, phosphoinositide 3-kinase; eNOS, endothelial nitric oxide synthase.

however was not affected by the same concentrations of cOCN (*Kondo et al., 2016*). Total OCN (>30.0 ng/mL) was reported to increase Akt and eNOS phosphorylation in human umbilical vein endothelial cells (HUVECs) in a time- and dose-dependent manner (*Dou et al., 2014*). Another study in HUVECs, in which insulin resistance was induced by tunicamycin, ucOCN (5 ng/mL) improved insulin signal transduction via PI3K/Akt/NF-κB pathways (*Guo et al., 2017*). Similarly, mouse vascular endothelial cells (VECs) and smooth muscle cells (VSMCs) were treated with 5 ng/mL ucOCN which protected against tunicamycin induced endoplasmic reticulum (ER) stress and autophagy, and improved insulin signalling in an Akt/mTOR/NF-κB pathway (*Zhou et al., 2013*).

*Idelevich, Rais & Monsonego-Ornan (2011)* importantly identified that OCN may indeed be an active player in vascular calcification, and not merely a product of differentiated VSMCs to an osteogenic phenotype. They demonstrated that overexpression of OCN stimulated glucose utilisation through activation of HIF-1α and promoted mineralisation and osteogenic differentiation in mouse vascular smooth muscle cells (MOVAS). Silencing of OCN RNA suppressed these effects. However, this does not distinguish an effect of osteoblast derived, circulating OCN, but rather locally produced effects from overexpression. However, when total OCN was exogenously added to wild-type MOVAS, OCN was shown to be a glucose metabolism-modulating factor, through HIF-1α. No
investigations have been conducted in human vascular smooth muscle cells to date. Ultimately, further investigation is needed into the potential direct influence of OCN on vascular calcification.

Overall, these studies support a hypothesis that OCN, ucOCN particularly, is directly vasoactive, and can increase nitric oxide via the PI3K/Akt/eNOS signalling pathway, which ultimately influences vasodilation. Additionally, further atheroprotective effects are demonstrated through protecting against high fatty acid induced apoptosis and improving insulin signalling. Although *Jung et al. (2013)* who used relatively physiological concentrations demonstrated a potential positive role of ucOCN in HAECs (increasing NO production and thus improving endothelial function), Idelevich et al. in contrast propose that OCN stimulates mineralisation and differentiation of mice VSMCs. It remains to be seen whether these disparities are due to differing effects of forms of OCN within different cell types and species, incubation timings or origin and concentrations of OCN, and whether OCN displays various functionalities that cannot be simply categorised into a good or bad role within the vasculature. The direct role of OCN within inflammation in the vasculature has also not been investigated. These limited studies and abundant questions signal there is need to clarify the role of physiologically relevant concentrations of OCN in human vascular cells. Moreover, the effects of both unOCN and cOCN, and potentially other OCN fragments, must be distinguished and deserves greater focus. Particularly as unOCN is more likely to be in circulation than fully non carboxylated OCN.

# FIBROBLAST GROWTH FACTOR 23

Osteocytes, which are the most abundant bone cell type, are the predominant secretors along with osteoblasts of fibroblast growth factor 23 (FGF23). FGF23 is the first described bone hormone and original phosphotonin (*Bonewald & Wacker, 2013*; *Bonnet, 2017*). FGF23 is a 251 amino acid protein in humans regulated at gene and protein level (*Bonewald & Wacker, 2013*). FGF23 is proteolytically cleaved upon secretion. Susceptibility to cleavage is reduced by O-glycosylation by polypeptide N-acetylgalactosaminyltransferase 3 which allows more intact, biologically active FGF23 to be secreted (approximately 32 kDa) (*Erben, 2017*). On the other hand, phosphorylation of FGF23 promotes cleavage resulting in C-terminus and N-terminus fragments, whose biological activity is unclear. Soluble or transmembrane α-Klotho is believed to be the cofactor required to allow for activation of FGF receptors by FGF23, though in some tissues FGF23 can have klotho independent effects (*Urakawa et al., 2006*; *Chen et al., 2018*; *Richter & Faul, 2018*). Within bone, the actions of locally produced FGF23 by osteocytes, as well as osteoblasts, have not been fully elucidated. Within mice, FGF23 may be a regulator of bone mineralisation in an autocrine/paracrine manner through inhibiting tissue non-specific alkaline phosphatase (TNAP). This results in increased pyrophosphate concentrations, likely in a klotho independent manner due to low concentrations of klotho within bone (*Murali et al., 2016*; *Erben, 2017*).

## Circulating concentrations of FGF23

FGF23 reaches its target organs through the circulation. Similar to OCN, there is complexity in measuring FGF23 as there are various presenting forms. Intact FGF23 (iFGF23) is the

most commonly used measurement, while C-terminus FGF23 (cFGF23) assays can also be used which measure both iFGF23 and cFGF23 (*Smith, 2014*). iFGF23 is also subject to *ex vivo* degradation and exhibits diurnal variation, while cFGF2 concentrations can be influenced by iron levels, fibrous dysplasia or familial tumoral calcinosis (*Wolf, Koch & Bregman, 2013*; *Smith, 2014*). In adults, mean cFGF23 has been reported at 49.0 RU/mL, and iFGF23 at 26.1 pg/mL, but high intra-individual variability exists which limits its application as a diagnostic or management tool (*Smith et al., 2012*). Paediatric reference values have been published for cFGF23 only (22.0–91.0 RU/ml) (*Fischer et al., 2012*). In patients with chronic kidney disease (CKD), circulating FGF23 is increased with decreasing renal function, and can reach up to 1000 times higher than normal levels (*Larsson et al., 2003*; *Fliser et al., 2007*; *Gutierrez et al., 2008*; *Isakova et al., 2011*).

## FGF23 and the vasculature

A number of epidemiological studies in humans have reported associations between increased circulating FGF23 concentrations and vascular dysfunction (*Mirza et al., 2009a*), vascular calcification (*Roos et al., 2008*; *Desjardins et al., 2012*) and increased risk of cardiovascular disease (*Dalal et al., 2011*). FGF23 concentrations have been positively correlated with left ventricular hypertrophy (LVH), which is particularly apparent in CKD patient cohorts (*Hsu & Wu, 2009*; *Mirza et al., 2009b*). Other studies have not shown an association between FGF23 and vascular calcification or atherosclerotic events in CKD patients (*Seiler et al., 2014*; *Sarmento-Dias et al., 2016*). A recent meta-analysis of prospective studies concluded that the relationship between FGF23 and cardiovascular disease risk is unlikely to be causal (*Marthi et al., 2018*). An established line of thought is that increased FGF23 may follow, as opposed to initiate, cardiovascular dysregulation (*Stohr et al., 2018*). It is likely that negative effects of high FGF23 are due to dysregulated phosphate metabolism. However, the research surrounding FGF23 is constantly evolving as the investigation into α-klotho dependent and independent effects continues, as well as the potential effects of locally produced FGF23 within the cardiovascular system. FGF23 levels have also been associated with inflammation in CKD patients and in other inflammatory disorders (*David, Francis & Babitt, 2017*). The following section reviews the direct effects of exogenously added FGF23 on vascular cells in order to provide insight into the possible bone-derived effects as FGF23 travels through the circulation to reach its canonical target organs (see Table 2).

### FGF23 in human and animal cells

In recent years, literature has emerged directed at elucidating the direct effects of FGF23 in the vasculature. In human aortic smooth muscle cells (HASMCs), 10 ng/mL of exogenous FGF23 has been shown to increase phosphorylation of ERK (extracellular signal-regulated kinase) after 10 min in serum-starved cells, which in turn increased production of Early Growth Response Protein-1 (EGR-1) (*Nakahara et al., 2016*). Also in human VSMCs, 10 ng/mL of FGF23 increased hydrogen peroxide production and did not induce nitric oxide (NO) production on its own, however when soluble klotho and phosphate were also added, NO was increased (*Six et al., 2014*). In the context of calcification, 5 ng/mL of

**Table 2** Summary of studies investigating *in vitro* effects of FGF23 on human and animal vascular cells.

| Study | Cell type | Context | Concentration | Results | Conclusions |
|---|---|---|---|---|---|
| *Lim et al. (2012)* | Human ASMCs | Calcification | 5 ng/mL | FGF23 stimulated proliferation and phosphorylation of ERK and AKT in a klotho dependent fashion. Pre-treatment with FGF23 and calcitriol followed by treatment with calcification media inhibited development of calcification, in a klotho dependent fashion. | Vitamin D receptor activators can restore Klotho expression and unmask FGF23 anti-calcific effects. Klotho is required for vascular FGF23 signalling. |
| *Zhu et al. (2013)* | Murine VSMCs | Calcification | 10 and 50 ng/mL | FGF23 reduced calcium deposition and increased phosphorylation of ERK1/2 (but not AKT). Concomitant exposure to FGF-23 and an ERK1/2 inhibitor (PD98059; 10 $\mu$M) increased calcification. | FGF23 signalling is likely to be a protective mechanism in calcification. |
| *Lindberg et al. (2013)* | Bovine VSMCs | Calcification | 0.125–2 ng/mL | FGF23 did not modify calcification. | No support for a role of FGF23 in vascular calcification. |
| *Scialla et al. (2013)* | Human VSMCs, murine VSMCs | Calcification | 1, 2, 10, 20 and 50 ng/mL | FGF23 had no effect on phosphate uptake or phosphate-induced calcification even in the presence of soluble klotho. FGF23 did not induce phosphorylation of ERK or FRS2$\alpha$. | No support for a role of FGF23 in vascular calcification |
| *Silswal et al. (2014)* | MurineAECs | Normal | 9 ng/mL | FGF23 increased superoxide levels compared with vehicle which was inhibited by pre-treatment with tiron. | FGF23 may reduce vasorelaxation by decreasing NO. |
| *Six et al. (2014)* | Human VSMCs, HUVECs | Normal | 10 ng/mL | $H_2O_2$ concentrations increased with FGF23 and klotho in HVSMCs but not in HUVECs. FGF-23 augmented $H_2O_2$ concentration induced by phosphate. FGF23 did not increase NO production in HUVECs, but FGF23, klotho and phosphate together, did. | Klotho deficiency may be deleterious as appears protective against increased ROS production from FGF23/phosphate. |
| *Nakahara et al. (2016)* | Human ASMCs | Normal | 10 ng/mL | FGF23 increased phosphorylated ERK, which subsequently increased EGR-1 expression. | FGF23 increased phosphorylated ERK expression and EGR-1 in HASMCs. |

**Table 2** (*continued*)

| Study | Cell type | Context | Concentration | Results | Conclusions |
|---|---|---|---|---|---|
| *Richter et al. (2016)* | Human CAECs | Normal | 10 ng/mL | FGF23 increased activation of FGFR1. Klotho levels were unaffected. FGF23 enhanced Klotho release and increased levels of ADAM17. FGF23 increased NO through Akt and eNOS. Cytosolic ROS increased via Nox2 and ROS detoxification via SOD2 and CAT. Blocking Klotho resulted in enhanced ROS formation and reduced NO availability. | Excess FGF23 may promote oxidative stress and endothelial dysfunction. |
| *Chung et al. (2017)* | Human AECs and Human BMECs | Normal | 0.5–100 ng/mL | FGF23 stimulated cell proliferation, eNOS, and NO production in AECs. High phosphate mitigated these effects which could not be rescued by soluble Klotho. Adding soluble α-Klotho following α-Klotho knockdown also did not rescue EC resistance to FGF23. None of these effects were observed in HBMECs. | FGF23 promoted proliferation and phosphorylation of eNOS. AECs were only responsive to FGF23 in the presence of α-Klotho synthesized by ECs. HBMECs did not respond to FGF23. |
| *Verkaik et al. (2018)* | Human MVECs | Normal | Approx. 3.2 ng/mL | 30 min treatment with FGF23 did not increase ERK phosphorylation. | FGF23 did not induce signal transduction through ERK. |

**Notes.**

Abbreviations: FGF-23, fibroblast growth factor 23; VSMCs, vascular smooth muscle cells; AECs, aortic endothelial cells; ASMCs, aortic smooth muscle cells; HUVECs, human umbilical vein endothelial cells; OPG, osteoprotogerin; MVECs, microvascular endothelial cells; BMECs, brain microvascular endothelial cells; CAECs, coronary artery endothelial cells; FRS2α, FGF receptor substrate 2α; EGR-1, early growth response protein-1; NO, nitric oxide; ECs, endothelial cells; eNOS, endothelial nitric oxide synthase; ROS, reactive oxygen species; Nox2, NADPH oxidase 2; SOD2, superoxide dismutase 2; CAT, catalase; FGFR1, fibroblast growth factor receptor 1.

FGF23 increased HASMC proliferation after 24 h in cells cultured in calcification media, and increased phosphorylation of ERK and AKT in a klotho-dependent manner (*Lim et al., 2012*). 24 h pre-treatment with FGF23 and calcitriol, followed by culture in calcification media, deterred calcification in HASMCs in a klotho dependent fashion, but this effect was not seen with FGF23 alone (without calcitriol) (*Lim et al., 2012*). *Scialla et al. (2013)* did not find any evidence for a pro-calcific effect of FGF23 in HASMCs (1–50 ng/mL), nor an increase in ERK phosphorylation (*Scialla et al., 2013*). Reflecting results shown in human VSMCs, FGF23 alone (10 and 50 ng/mL) reduced calcium deposition in the context of calcification in murine VSMCs through ERK signalling (*Zhu et al., 2013*), whereas in bovine VSMCs 0.2–2.0 ng/mL FGF23 did not modify calcification (*Lindberg et al., 2013*). In a study with murine VSMCs no effect of FGF23 on calcification was observed (1–50 ng/mL) (*Scialla et al., 2013*).

In microvascular endothelial cells, FGF23 did not increase ERK phosphorylation, nor in human umbilical vein endothelial cells did FGF23 increase hydrogen peroxide production (*Six et al., 2014; Verkaik et al., 2018*). In HAECs, FGF23 (50 and 100 ng/mL) increased proliferation after 48 h, while 10 ng/mL stimulated phosphorylated eNOS and NO production (*Chung et al., 2017*). These results were no longer observed following α-klotho knockdown and could not be rescued by addition of soluble klotho, indicating

that these effects were dependent on localised endogenous α-klotho production in HAECs. Interestingly, no increase in proliferation, nor eNOS phosphorylation or NO production were observed in the same conditions in human brain microvascular cells (*Chung et al., 2017*). In coronary artery endothelial cells (HCAECs), *Richter et al. (2016)* demonstrated that 10 ng/mL of FGF23 increased expression of the FGF receptor-1, increased eNOS phosphorylation and NO in an Akt-dependent manner, increased ROS production and ROS detoxification, and additionally increased secretion of soluble klotho via upregulation of ADAM17 (*Richter et al., 2016*). In blocking klotho, the authors further demonstrated an upregulation of ROS and decrease in NO. In murine aortic endothelial cells, 9 ng/mL FGF23 significantly increased superoxide levels (*Silswal et al., 2014*).

These collated results begin to untie whether FGF23 may have any causal effects within the context of cardiovascular pathologies. A few conclusions can be made including: FGF23 initiates direct effects on vascular cells including SMCs and ECs but actions may be tissue specific; FGF23 appears to increase eNOS phosphorylation and NO production likely in an Akt dependent manner, and predominantly in a klotho-dependent manner (perhaps restricted to locally produced klotho and not systemic circulating klotho); however, FGF23 also appears to increase ROS and oxidative stress, which is counteracted by klotho; within calcification models, FGF23 increased proliferation of SMCs, but does not appear to increase calcification and indeed may be anti-calcific. Bringing these results together paints a conflicting but potential positive role of FGF23 in vascular function. The results presented encourages the hypothesis that the associations and correlations between FGF23 and CVD are most likely due to deleterious effects of altered phosphate and mineral metabolism and indeed that FGF23 levels likely follow, as oppose to direct, cardiovascular disturbances. It is a complex area, with further discrepancies such as variable physiological and disease states, deficiency of klotho, origin of secretion of FGF23 and klotho, and many more factors leaving the role of FGF23 in the vasculature not a simple question to be answered. Further confirmative examinations are necessary and investigations into whether N- and C-terminal fragments of FGF23 are additionally biologically active and if so, how they differ from intact FGF23. Interestingly, any direct pro- or anti-inflammatory properties of FGF23 on vascular cells have not yet been explored despite accumulating evidence in other tissues for a role of FGF23 in inflammation (*David, Francis & Babitt, 2017*).

## LIPOCALIN-2

Lipocalin-2 (LCN-2), also referred to as neutrophil gelatinase-associated lipocalin (NGAL) or 24p3, is the most recently identified osteokine. However, it is debatable whether LCN-2 can be strictly described as such as it is secreted by a wide variety of cell types, and has indeed been more widely described as an adipokine to date. Despite this, it has recently been reported to be produced in bone ten-fold more than in white fat, at least in mice, and it remains to be reported whether this is also the case in humans (*Mosialou et al., 2017*). As such, LCN-2 is predominantly an osteoblast and adipocyte derived 25 kDa secreted glycoprotein which acts as a lipid chaperone positioning itself as a key pro-inflammatory link between obesity and associated metabolic disorders and vascular disease (*Kjeldsen*

*et al., 1993*; *Wang, 2012*). It exists *in vivo* as a monomer, homodimer, or heterodimer by forming a complex with matrix metalloproteinase-9 (MMP-9) which stabilizes MMP-9 by preventing autodegradation (*Kjeldsen et al., 1993*; *Yan et al., 2001*; *Chakraborty et al., 2012*). LCN-2 has been identified to mediate an innate immune response to bacteria by sequestering iron and induces apoptosis in many cell types (*Devireddy et al., 2001*; *Flo et al., 2004*).

## Circulating concentrations of LCN-2

LCN-2 is increased in obese individuals, which is presumed derived from increased adipose tissue and also increased expression from the liver, but may also be due to changes in skeletal homeostasis which is also altered in obesity (*Wang et al., 2007*; *Luo et al., 2016*). It has been suggested that LCN-2 can be used as a biomarker for early renal injury (*Mishra et al., 2004*; *Mishra et al., 2005*). For example, in children undergoing surgery, those who did not subsequently develop acute renal injury had LCN-2 serum concentrations <50 ng/mL, but 50% of those who did develop acute renal injury has concentrations >50 ng/mL (*Mishra et al., 2005*). In a study of 1,203 Chinese obese and non-obese men, LCN-2 serum concentrations were in the range 29.3–53.5 ng/mL (*Luo et al., 2016*). In patients with Type 2 Diabetes, serum levels were significantly higher in those with subclinical atherosclerosis than those without (112.9 ng/mL versus 77.2 ng/mL) (*Xiao et al., 2013*). LCN-2 serum concentrations are also higher in patients with metabolic syndrome compared to those without (83.2 ng/mL versus 67.5 ng/mL) (*Wang et al., 2007*).

## Lipocalcin-2 and the vasculature

LCN-2 is a novel, debatable osteokine. Examining its effects on vascular cells when originating from osteoblasts presents a difficult task, requiring for example tissue specific knock-down studies. Further validation to characterise LCN-2 as an osteokine with more convincing human data is primarily needed. As LCN-2 has been associated with a stage-dependent contribution to atherosclerosis, promoting lipotoxicity within the vasculature in obese states, and causing endothelial dysfunction and cardiovascular complications, it is important to establish whether bone-derived LCN-2 is a key mediator that could be therapeutically targeted (*Wang, 2012*; *Amersfoort et al., 2018*). LCN-2 is also proposed as a sensitive marker for cardio-renal disease in patients with acute heart failure (*Mishra et al., 2005*; *Alvelos et al., 2011*).

Most studies to date have focused on observational and epidemiological data on the involvement of LCN-2 in obesity and diabetic associated vascular complications. It is currently unfounded to speculate whether these effects are influenced by bone derived LCN-2. The limited investigations on the direct exogenous addition of this interesting protein in vascular cells are summarised below and in Table 3.

### LCN-2 in human and animal cells

Wang and colleagues, interested in investigating pulmonary hypertension, demonstrated that in human pulmonary artery smooth muscle cells (HPASMCs), LCN-2 at 10 and 20 ng/mL increased proliferation after 24 hrs (*Wang et al., 2015*). LCN-2 promoted the activity of the PI3-K pathway by increasing Akt phosphorylation (3–30 ng/mL) after 24

**Table 3  Summary of studies investigating *in vitro* effects of LCN-2 on human and animal vascular cells.**

| Study | Cell type | Concentration | Results | Conclusions |
|---|---|---|---|---|
| *Lee et al. (2011)* | Rat BMVECs | 10,000 ng/mL | LCN-2 increased expression of chemokine CXCL10 | LCN-2 may promote chemoattractants and neuroinflammation |
| *Wang et al. (2014)* | Human PASMCs | 10 ng/mL3–100 ng/mL | Decreased serum deprivation induced apoptosis and $H_2O_2$ induced apoptosis with LCN-2. LCN-2 decreased the cleavage and activity of caspase-3, and expression of Bax (apoptotic protein). LCN-2 increased expression of SOD1/2 and decreased intracellular ROS. | LCN-2 appears protective against apoptosis and decreased ROS |
| *Wu et al. (2015)* | Rat BECs | 500–2,000 ng/mL | Enhanced Matrigel tube formation with LCN-2 and migration of cells via iron and ROS related pathways | LCN-2 may contribute to neurovascular recovery |
| *Wang et al. (2015)* | Human PASMCs | 3–30 ng/mL | LCN-2 increased proliferation, at least in part via PI3-K signalling pathway | LCN-2 increased HPASMC proliferation |
| *Eilenberg et al. (2016)* | HUVECs, Human CASMCs | 200 ng/mL, 500 ng/mL or 1,000 ng/mL | Increased secretion of inflammatory markers IL-8, IL-6 and MCP-1 dose-dependently with LCN-2 | LCN-2 may be pro-inflammatory |
| *Wang et al. (2017)* | Human PASMCs | 10 ng/mL | LCN-2 increased ER stress and proliferation via increased intracellular iron levels | LCN-2 increased HPASMC proliferation and ER stress |

**Notes.**

Abbreviations: LCN-2, lipocalin-2; BMVECs, brain microvascular endothelial cells; BECs, brain endothelial cells; HUVECs, human umbilical vein endothelial cells; CASMCs, coronary artery smooth muscle cells; PASMCs, pulmonary artery smooth muscle cells; ROS, reactive oxygen species; IL-8, interleukin-8; IL-6, interleukin-6; MCP-1, monocyte chemoattractant protein-1; SOD1/2, superoxide dismutase 1/2; ER, endoplasmic reticulum.

hrs, which was abrogated by an Akt inhibitor (*Wang et al., 2015*). This inhibitor also partly prevented the LCN-2 induced increase in proliferation. Interestingly, this group previously showed in the same cell type that 10 ng/mL LCN-2 decreased serum deprivation induced apoptosis after 24 hrs compared to control, and furthermore decreased the susceptibility of HPASMCs to $H_2O_2$ induced apoptosis (*Wang et al., 2014*). They further showed that LCN-2 was decreasing the cleavage and activity of caspase-3, and decreasing expression of Bax, an important pro-apoptotic factor. Finally, they confirmed LCN-2 (3–100 ng/mL for 24 hrs) increased the expression of superoxide dismutase 1 and 2 (SOD1 and SOD2) dose-dependently and 10 ng/mL decreased intracellular reactive oxygen species (ROS) both with and without $H_2O_2$. This group went on to show that LCN-2 (10 ng/mL) in fact promoted endoplasmic reticulum (ER) stress and proliferation within HPASMCs via increased intracellular iron levels (*Wang et al., 2017*).

HUVECs and human coronary artery smooth muscle cells (HCASMCs) treated with LCN-2 (200 ng/mL, 500 ng/mL or 1μg/mL) significantly increased secretion of inflammatory markers interleukin-8 (IL-8), IL-6 and monocyte chemoattractant protein-1 (MCP-1) in a dose-dependent manner (*Eilenberg et al., 2016*). Within the central nervous

system, LCN-2 (10 µg/mL) was shown to increase mRNA expression of C-X-C motif chemokine 10 in mice brain microvascular endothelial cells (BMVECs), providing preliminary evidence that LCN-2 may act as a chemoattractant inducer and promote neuroinflammation (*Lee et al., 2011*). In another study in rat brain endothelial cells, LCN-2 (0.5–2 µg/mL) enhanced angiogenesis, shown through Matrigel tube formation and scratch migration, via iron and ROS related pathways (*Wu et al., 2015*).

Despite the clinical associations between LCN-2 and vascular disturbances, the mechanisms at a cellular level by physiological concentrations of exogenous LCN-2 have been largely disregarded and research is warranted. The investigations to date have largely focused on LCN-2 in the context of pulmonary hypertension and the use of HPASMCs. Within this cell type LCN-2 is shown to have anti-apoptotic and pro-proliferative effects, while also decreasing ROS but promoting ER stress. It has also been demonstrated that LCN-2 may be pro-inflammatory in a number of cells types, which may be viewed as beneficial in the acute inflammatory response (*Castellheim et al., 2009*). Concentrations of LCN-2 used in the studies reviewed have been quite varied (3 ng/mL –10 µg/mL). It is clear there is a large scope of research yet to be completed in this area to inform researchers and clinicians about any potential contribution of LCN-2 in vascular physiological and pathophysiology.

## SUMMARY AND CONCLUSIONS

The effects of bone-derived factors on systemic health is an emerging, interesting branch of endocrinology research. As conflicting epidemiological reports have linked these bone-derived factors with multiple vascular pathologies such as atherosclerosis and calcification, there is interest in their therapeutic target potential. In order to derive any causal effects, it is important to review the basic *in vitro* science that has been carried out to date. Thus, this review aimed to collate studies investigating the *in vitro* effects of OCN, FGF23 and LCN-2 to clarify whether there is support for a basic biological or pathological role within the vasculature. Overall, the evidence base points to direct vasoactive properties of the osteokines OCN, FGF23 and LCN-2 across human and animal cells. Both OCN and FGF23 have been demonstrated to be anti-apoptotic and increase eNOS phosphorylation and NO production through Akt signalling. Taken together, these findings suggest a role for these osteokines in supporting normal vascular functionality. OCN has been implicated in improving insulin signalling, and ensuing protectiveness against ER stress. OCN may be involved in calcification but further research is warranted, while there is no evidence at least of a pro-calcific effect of FGF23 *in vitro*. FGF23 and LCN-2 have been shown to increase proliferation in some cell types and increase and decrease ROS respectively. LCN-2 has been shown to have anti-apoptotic effects and may increase ER stress as well as have pro-inflammatory properties. LCN-2 has also been demonstrated to promote angiogenesis. There is no strong evidence to suggest a pathological role of OCN or FGF23 in the vasculature, however further research with LCN-2 needs to be completed in order to better understand its role. Despite the topical area, a number of preliminary questions remain to be answered. For example, the role of OCN and FGF-23 in inflammation, the

role of OCN and LCN-2 in calcification, as well as their cellular receptors and related downstream signalling pathways in vascular cells. Various cleaved forms of the circulating proteins should also be investigated to assess their biological activity. It is clear that further, human data should be generated to understand the roles and mechanisms of actions of these osteokines in physiological and pathophysiological states.

### Funding
This work was supported by the Biotechnology and Biological Sciences Research Council [grant number BB/M008770/1]. The funders had no role in study design, data collection and analysis, decision to publish, or preparation of the manuscript.

### Grant Disclosures
The following grant information was disclosed by the authors:
Biotechnology and Biological Sciences Research Council: BB/M008770/1.

### Competing Interests
The authors declare there are no competing interests.

### Author Contributions
- Sophie A. Millar conceived and designed the experiments, performed the experiments, analyzed the data, prepared figures and/or tables, authored or reviewed drafts of the paper, approved the final draft.
- Susan I. Anderson and Saoirse E. O'Sullivan authored or reviewed drafts of the paper, approved the final draft.

### Data Availability
This was a literature review.

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
