# Peer review of "Osteokines and the vasculature: a review of the *in vitro* effects of osteocalcin, fibroblast growth factor-23 and lipocalin-2"

_PeerJ, doi:10.7717/peerj.7139_

## Round 0.1 · original submission · Major Revisions

Three expert reviewers have evaluated the manuscript. They agreed, that the review topic is interesting for the scientific community, but amendments are needed. I support the opinion, that the paper need to be better focused and it should be emphasized why this review is different from many previous ones in the literature.

Reviewer 1 ·

Basic reporting

- The tables are a good idea and makes it very easy to interpret each study, however, in general the results and conclusion sections are too long, they should be a summary of what occurs in each study, otherwise it is no different to reading about the study in the main text.
- In the Tables, please ensure a consistent measure of concentration is chosen (for example: ng/ml), to assist the reader in interpreting the differences between the studies.

Experimental design

- Too much background information is provided on each osteokine in comparison to the discussion on each osteokine and the vasculature effects. A suggestion would be to remove the subheadings ‘circulating concentrations of…., expression of ….. and endocrine functions of…) and only include background information that is needed to discuss and interpret the vasculature effects.
- Given that this is not a systematic review why have you included a section on the search strategy? If it was review systematically then that will need to be reflected in the title and follow the appropriate guidelines for systematic reviews.
- At the end of the introduction the aims of the article include a section on the endocrine functions of each osteokine, however, the title for this review is ‘osteokines and the vasculature’ as you pointed out in the text, recent reviews have been completed on the endocrine functions of OCN and FGF-23. It may be beneficial to rethink how much information should be included on the endocrine functions of each osteokine and what is novel about this review, compared to previous ones.

Validity of the findings

- Please describe what is novel about the section on OCN and the vasculature. A recent review ‘Tacey, A., Qaradakhi, T., Brennan-Speranza, T., Hayes, A., Zulli, A., & Levinger, I. (2018). Potential Role for Osteocalcin in the Development of Atherosclerosis and Blood Vessel Disease. Nutrients, 10(10), 1426’ described the effect of OCN on atherosclerosis development and included all of the studies that were included in your discussion. As there have not been any new studies published since the previous review, what is this section adding to the body of knowledge that was not discussed in the recent review?
- In the summary and conclusion section, line 492 – 493 you describe that the evidence points towards direct vasoactive properties of the osteokines in human and animal cells. However, in the FGF-23 section you describe in lines 376 - 379 ‘The results presented encourages the hypothesis that the associations and correlations between FGF23 and CVD are most likely due to deleterious effects of altered phosphate and mineral metabolism and indeed that FGF23 levels likely follow, as oppose to direct, cardiovascular disturbances.’
- Several of the studies that are referred to, especially in the FGF-23 and LCN-2 sections involve microvascular vessels (some examples on lines – 352, 360, 467), yet the focus throughout the text is on atherosclerosis and calcification, which are macrovascular diseases. You may need to clarify whether you are looking at one/both micro- and macro-vascular diseases and adjust the included articles accordingly.

Additional comments

- in the introduction there is too much background on the skeleton and blood vessels as a whole and not enough specificity on what you are discussing. The 3 osteokines you are discussing are only mentioned in ~ 2 sentences of the whole introduction. Further, things that you are not discussing in the main text (osteopontin, osteoprotegerin, sclerostin and DKK1) do not need to be discussed in the introduction. The introduction needs to be shortened and more focused, describing more background and rationale for the review.
- lines 125 – 128 – There is several mistakes in the reporting of the different forms of OCN, please refer to ‘Li, J.; Zhang, H.; Yang, C.; Li, Y.; Dai, Z. An overview of osteocalcin progress. J. Bone Miner. Metab. 2016, 34,367–379’.

·

Basic reporting

The review article by Millar et al. summarizes the endocrine functions and in vitro effects of osteokines on vascular cells. This review article is clearly written and well organized with relevant subheadings. Tables are comprehensive and helpful. The review is cross-disciplinary and will be of interest to a wide audience including life science and medical researchers as well as clinicians. Some minor comments are stated below in the “Comments for the author” section.

Experimental design

The search strategy used to retrieve articles related to this literature review is excellent with search encompassing multiple databases. The review article is logically organized into relevant subsections for all the three osteokines described in the text.

Validity of the findings

The conclusions are well supported by the background and existing literature on osteokines. Furthermore, some knowledge gap existing in this field has also been highlighted well towards the end of conclusions.

Additional comments

Some minor comments and suggestions are as follows:
1. Lines 62, 93 & 96: Osteoprotegerin is misspelled (it should be ‘e’ instead of ‘o’ after ‘t’)
2. Line 63: RANKL should not have a hyphen between K & L.
3. Lines 69 & 95- use a semicolon to separate multiple references and order them chronologically as stated in the “Instructions for authors” submission guidelines on PeerJ website. This needs to be checked throughout the article.
4. Line 76: Abbreviate cardiovascular disease as CVD in addition to writing the full-form (as CVD has been used later in the text in line 377).
5. Line 122: The full-length osteocalcin (OCN) is 95 and 100 amino acid protein in mice and humans respectively. There is a signal peptide of 22 amino acids and a propeptide of 28 amino acids in humans. It’s the mature protein that has 49 amino acids in humans (see Chapter 2- Bone, matrix and mineralization-Pediatric bone-2nd edition-Marc D. McKee, William G. Cole 2012 OR search protein database in NCBI for full length protein sequence). So to say “OCN is 46 and 49 amino acid protein in mice and humans” as stated in this review is not fully correct. Either say “the mature OCN is 46 and 49 ….” OR state “the full-length OCN is 95 and 100… ”
6. Line 145: “fragment from proteolytic enzymes” sentence needs to be corrected as “fragments released by proteolytic cleavage” (the fragments are not from proteolytic enzymes but are from osteocalcin itself.)
7. Line 151: Hannemann is misspelled (it should be ‘nn’ after ‘Ha’ instead of ‘mm’).
8. Line 163: The suitable word is “identified” instead of “expressed”.
9. Line 241: It should be “suppressed these effects” instead of “suppressed these results”.
10. Lines 260-261: “deserve” instead of “deserves”.
11. Line 266: Change “251 amino acid peptide” to “251 amino acid protein” (its big enough to be called as a protein, in general less than 50 amino acids is designated as a peptide).
12. Line 268: “N-acetylgalactosaminyltransferase” (delete the space between galactosa and minyl)
13. Line 284: It should be “measure” instead of “measures” (as the preceding words say C-terminus FGF23 assays, correct one or the other).
14. Line 335: Write also the full form of ERK as extracellular signal-regulated kinase (as ERK has been used later at multiple places, so would be useful to write its full form here)
15. Lines 338-339: Change “when soluble klotho and phosphate was” to “….were”.
16. Lines 397-398: It should read as “monomer, homodimer, or heterodimer by forming a complex with matrix metalloproteinase-9 (MMP-9)”. Also, LCN-2 stabilizes MMP-9 by preventing MMP-9 from autodegradation rather than increasing its activity. (See Chakraborty et al., Biochim Biophys Acta., 2012.) So to say LCN-2 increases its activity as in line 398 is rather conflicting.
17. Line 424: Change “though” to “through”.
18. Line 430: The full form of MC4R should be written (melanocortin 4 receptor).
19. Line 433: “present” instead of “presents”
20. Lines 455 & 460: H2O2 – It appears there is a 0 (zero) mistakenly typed there instead of O (capital letter O).
21. Line 467: CXCL10 should also be stated in its full form as C-X-C motif chemokine 10.
22. Line 501 & 32: It should read as “have” instead of “having”.
23. Table 2: Legend has “FGF-23”.It should be written as FGF23 just to be consistent with the rest of the text in the review. Another minor correction- enter the units of concentration in the last entry of table 2 - “Richter et al., 2016” which is “10 ng/ml” in the concentration column.

Reviewer 3 ·

Basic reporting

In this review, the authors introduced and discussed the in vitro effects of OCN, FGF23 and LCN2. They focused on the function of these genes in vasculature including the effects on vascular-related cells. In general, this review was well written with sufficient information and discussion. I have only some suggestions:

1. LCN2 plays an important role in neuroinflammation as well as blood brain barrier. Currently the authors only mentioned these points in a pretty simple way. Could the authors talk more about this since blood brain barrier is related to the topic of vasculature? It should attract more researchers working in these fields.

2. Could the authors make a table and put OCN, FGF23 and LCN2 side by side to summarize the expression pattern of these three genes?

Experimental design

no comments

Validity of the findings

no comments

Additional comments

no comments

---

## Round 0.2 · Minor Revisions

The manuscript has been substantially improved. I ask the authors to amend the manuscript based on the remaining minor comments of Reviewer #1.

Reviewer 1 ·

Basic reporting

- You describe that there is conflicting observational data on these osteokines and vasculature interaction in Lines 74-75. It may be worthwhile including some references here to support this statement and identify the rationale for investigating the osteokine and vasculature interaction.
- The professionalism of the writing needs to be improved throughout to make it acceptable for publication in this journal, some suggested changes are listed below.
o Line 40 – ‘our bones’ should be changed to something like bone or the skeleton
o Line 43 – remove ‘the’
o Several sentences are too long and confusing, some examples are; lines 100 – 103, lines 109 – 112 and lines 200 – 204. They should be split into several sentences or shortened.
o Line 109 – what is meant by ‘its abundance’? are you referring to OCN or ucOC?
o Line 251 – 252 – did not find repeated twice
o line 323 – 325 – the sentence needs to be made clearer
- The use of abbreviations is not consistent throughout document, for example osteocalcin is written in full on lines 105 + 109, after the acronym has been used.
- The abbreviation for undercarboxylated osteocalcin is different in the text (line 102) compared to in Table 1 – this will need to be fixed

Experimental design

- The review is nicely separated into coherent subsections and has a logical flow.

Validity of the findings

no comment

Additional comments

- In the reporting of uncarboxylated and undercarboxylated osteocalcin the authors should be aware that the form found in circulation is likely undercarboxylated osteocalcin. Whereas fully uncarboxylated osteocalcin likely does not occur endogenously and is only a synthetic compound. You have reported throughout the text that uncarboxylated osteocalcin is the form found in circulation.

·

Basic reporting

The authors have addressed the concerns highlighted in the previous review. The review reads clear now and is within the scope of the journal.

Experimental design

No comment

Validity of the findings

No comment

Additional comments

No comment

Reviewer 3 ·

Basic reporting

I thank the authors for their efforts to make the manuscript look better. I have no more questions.

Experimental design

No comments

Validity of the findings

No comments

Additional comments

No comments

---

## Round 0.3 · accepted · Accept

All the comments were adequately addressed during the second revision.